# CalibRBEV: Multi-Camera Calibration via Reversed Bird's-eye-view Representations for Autonomous Driving

## ABSTRACT

Camera calibration consists of determining the intrinsic and extrinsic parameters of an imaging system, which forms the fundamental basis for various computer vision tasks and applications, *e.g.*, robotics and autonomous driving (AD). However, prevailing camera calibration models pose a time-consuming and labor-intensive off-board process particularly in mass production settings, while simultaneously lacking exploration of real-world autonomous driving scenarios. To this end, in this paper, inspired by recent advancements in bird's-eye-view (BEV) perception models, we proposes a novel automatic multi-camera **Calib**ration method via **R**eversed **BEV** representations for autonomous driving, termed **CalibRBEV**. Specifically, the proposed CalibRBEV model primarily comprises two stages. Initially, we innovatively reverse the BEV perception pipeline, reconstructing bounding boxes through an attention autoencoder module to fully extract the latent reversed BEV representations. Subsequently, the obtained representations from encoder are interacted with the surrounding multi-view image features for further refinement and calibration parameters prediction. Extensive experimental results on nuScenes and Waymo datasets validate the effectiveness of our proposed model.

## CCS CONCEPTS

• **Computing methodologies** → **Multimedia Interpretation**; • **Multimedia Interpretation** → Camera calibration.

## KEYWORDS

Multi-camera calibration, bird's-eye-view (BEV), autonomous driving (AD).

## 1 INTRODUCTION

Camera calibration is a fundamental procedure in computer vision tasks [43, 52], mainly involving the determination of intrinsic and extrinsic parameters of an imaging system. Specifically, these parameters are applied to the original images to obtain spatial coordinates in the real world before downstream tasks such as object detection or segmentation.

In modern autonomous driving (AD) systems, vehicles are typically outfitted with multiple sensors (*e.g.*, multi-cameras, LiDAR). Thus, employing classic and popular calibration methods [41, 61]

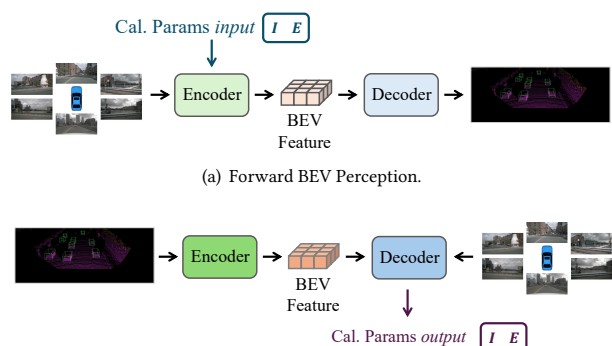

(a) Forward BEV Perception.

(b) Reversed BEV Multi-Camera Calibration.

**Figure 1: The difference between forward BEV perception pipeline and our proposed reversed BEV calibration pipeline. For simplicity, the process of extracting multi-view image features using backbone network is omitted here. (a) Forward BEV Perception. Multi-view images and calibration parameters are utilized by the encoder to generate BEV features, which are then forwarded through the decoder head to yield predictive bounding boxes. (b) Reversed BEV Calibration. The model firstly encodes bounding boxes into BEV features, which are subsequently combined with multi-camera images to predict the calibration parameters.**

such as hand-eye calibration is a time-consuming and laborious off-board procedure during mass production, let alone re-calibration after product delivery. For example, for the extrinsic parameter matrix, it usually requires to be executed within a specific calibration room. On the other hand, some recent deep learning-based calibration models achieve performance potential by directly regressing calibration parameter targets [31, 54]. However, there is a notable dearth of comprehensive exploration within the realm of multi-view cameras complex scenes in autonomous driving.

Recent bird's-eye-view (BEV) models [34, 35] achieve notable progress in 3D perception using multi-camera imagery and calibration parameters within the context of AD scenarios. As shown in Fig. 1(a), the encoder utilizes multi-view images and calibration parameters to produce BEV representations, which are subsequently fed into the decoder to generate predictive bounding boxes, enabling perception in the real world. On one hand, such a perceptual paradigm necessitates accurate calibration parameters to achieve precise detection, highlighting the crucial role of camera calibration in AD. On the other hand, the close relationship between BEV models and calibration parameters inevitably raises a question: *can the outstanding BEV model concept be leveraged to accomplish the task of camera calibration in AD?*

Motivated by this observation, *we rethink the camera calibration from a new perspective, i.e., a reversed BEV perception pipeline.* In other words, reversely, our objective is centered on the optimization of calibration parameters utilizing bounding boxes data, rather than the prediction of bounding boxes, as shown in Fig. 1(b).

During model training, the multi-view images and bounding boxes can be readily acquired using alternate sensors from calibration datasets, such as cameras and high-precision LiDAR. Thus, the key challenges are *(i) The training of the whole model still demands a substantial pre-collect calibration dataset*, which goes against our initial intention of cost reduction. In the meanwhile, *(ii) it is still unexplored in the community that how high-quality and comprehensive BEV features can be extracted from bounding boxes, which is a reversal of existing literatures [34, 35].*

Regarding the above two issues, we employ an auto-encoder reconstruction strategy to pre-acquire the reversed BEV representations. Consequently, the representations derived from the encoding of 3D object bounding boxes should be able to capture comprehensive environmental information, facilitating accurate prediction of camera calibration parameters. Additionally, pre-training of the auto-encoder module for reconstruction enhances the robust generalization of the encoder, further diminish the requirement for extensive calibration datasets, as shown in Table 4.

To this end, this paper proposes the CalibRBEV model, which achieves automatic multi-camera calibration via reversed BEV representations for autonomous driving. Specifically, our model begins by reverse BEV perception pipeline and utilizes attention auto-encoder modules to reconstruct bounding boxes for obtaining the reversed BEV representations. Subsequently, these features interact with multi-view image features for further refinement to predict the final calibration parameters. Experimental results on nuScenes and Waymo datasets validate the effectiveness of our method. Particularly, on nuScenes dataset, we achieve 0.0102, 0.1637, and 0.0287 errors on translation vector, rotation matrix, and intrinsic parameters, respectively.

The main contributions of our work are summarized as:

- This work introduces the concept of BEV perception models into calibration tasks, achieving the camera calibration in autonomous driving scenarios through the reversed BEV pipeline.
- This paper proposes an auto-encoder-based strategy for reconstructing bounding box data to achieve high-quality reversed BEV feature extraction, thereby further enhancing the prediction of calibration parameters.
- Experimental results on nuScenes and Waymo datasets validate the effectiveness of our model. The ablation results further substantiate the performance of module design and its efficacy across varying data size.

## 2 RELATED WORK

In this section, we offer an extensive overview of related work concerning our proposed model, primarily focusing on three aspects: camera calibration, bird's-eye-view (BEV) perception models, and autonomous driving scenarios.

### 2.1 Camera Calibration

Camera calibration can be broadly categorized into two main directions [36, 44]: traditional hand-crafted methods and deep learning-based models. This subsection will delve into each direction separately.

*2.1.1 Traditional Models.* In traditional camera calibration models, calibration generally involves using established mathematical models and algorithms [1, 7, 12, 16, 19, 20, 41, 45, 61] to estimate camera parameters, which can be further categorized into three main categories [36], namely, calibration-plane-based models [2, 19, 45, 61], geometric-prior-based models [1, 7], and self-calibration models [15, 16, 20, 59].

(i) The calibration-plane-based models typically necessitate the use of standard references (*e.g.*, calibration target: checkerboard) in the world coordinate system to estimate parameters. A simple calibration pattern comprising equally spaced dots is proposed [45] for fish-eye lens camera. Zhang *et al.* [61] further introduced a simple and effective camera calibration model, wherein parameters are estimated through the utilization of key points from different viewpoints on the calibration plane. (ii) The geometric-prior-based models usually depend on geometric priors and properties for the prediction of calibration parameters. For example, the projective invariant properties [1] is employed in central catadioptric systems, while user-specified salient scene regions [7] are preserved according to human vision tendencies. (iii) The self-calibration models mainly directly take multiple images captured from different camera orientations to estimate the parameters. The epipolar structure of image pairs is leveraged in [16] and image correspondences is utilized in [20]. LiDAR edges information [59] is further used for automatic extrinsic calibration.

In general, these traditional calibration methods heavily rely on known scenes and mathematical models, which greatly limit their ability to generalize in complex scenarios. Furthermore, they necessitate the design of specific calibration plane and extensive manual intervention, inevitably adding to costs and proving unsuitable for mass production scenarios.

*2.1.2 Learning-based Models.* Recently, with the advancement of deep learning [14, 21, 32], a novel learning-based calibration method has emerged, demonstrating promising potential as it directly utilizes deep neural networks (DNNs) to fit the calibration targets. Following the [36] and considering the camera model, deep learning-based calibration methods are categorized into four groups: standard model [9, 31, 50, 54], distortion model [3, 8, 17, 60], cross-view model [13, 37, 57], and cross-sensor model [18, 46, 62, 64].

(i) The standard model aims to estimate the intrinsic and extrinsic parameters of the camera. Initially, DeepFocal [54] presented pioneering learning-based work to predict the focal length of cameras. The extrinsic parameters are similarly obtained through regression and are commonly regarded as camera pose estimation [9, 31]. Further techniques [4, 22, 50] have been suggested for the simultaneous estimation of both intrinsic and extrinsic parameters. (ii) The distortion model primarily focuses on calibrating radial distortion and roll shutter distortion. DeepCalib [3] firstly employed DNNs to regressively predict the camera distortion parameters. Later, more robust networks and training strategies are utilized to improve

calibration performance [8, 17]. (iii) The cross-view model extends the single-camera settings to the more intricate multi-camera calibration scenarios, such as homography matrix prediction [13] and multi-calibration parameters prediction [37, 57]. (iv) The cross-sensor model aims to achieve multi-sensor calibration, primarily aligning camera/LiDAR coordinates [18, 42, 62, 64].

However, despite the potential exhibited by the aforementioned learning-based camera calibration methods across diverse scenarios, further exploration is required in the challenging scenarios of autonomous driving. In this paper, we innovatively introduce the concept of BEV perception and incorporate reversed BEV features to directly predict the multi-view camera parameters in real-world autonomous driving scenarios.

## 2.2 BEV Perception

The BEV perception models [33] have garnered widespread attention in the research community for its benefits derived from a unified perspective and integration of multi-view or multi-sensor information, particularly in the context of AD scenarios. It is advisable to refer to the comprehensive review [33] to gain a comprehensive understanding of BEV perception models. Among these methods, perception strategies based on transformers have demonstrated outstanding performance [14, 49].

Initially, DETR [6] pioneered the utilization of the transformer architecture for the 2D object detection task. Given that cross-attention entails prolonged training periods, Deformable DETR [63] further enhances model efficiency by integrating deformable attention mechanisms. Inspired by the aforementioned 2D detection models, DETR3D [52] is a classic model that generalizes BEV method from 2D to 3D detection based on DETR and Deformable DETR.

Following the DETR3D, BEV models [25, 35, 38, 39] have experienced rapid development. PETR [38] and BEVDet [25] further improve the performance based on DETR3D. Then, BEVFormer [35], PETRv2 [39], BEVDet4D [24], and BEVDepth [34] exploit temporal features in multi-camera 3D object detection, and achieve significant improvements over single-frame methods. Moreover, BEVFusion [40] and FUTR3D [11] further fuse multi-modal features of various sensors such as camera and LiDAR into BEV features to boost the performance.

Our primary focus lies within methods pertaining to multi-view images BEV perception models, such as BEVFormer [35]. To ensure accurate detection, the utilization of multiple camera calibration parameters is imperative for multi-image inputs. In this paper, unlike above mentioned forward BEV perception models, we innovatively reverse the BEV pipeline to accomplish the camera calibration task.

## 2.3 Autonomous Driving

Modern autonomous driving (AD) systems are usually divided into several modules including perception [10, 33], prediction [27–29], planning [48], and etc. Though they have the advantages of decoupled developments and explainability, module-based systems suffer from cumulative errors and inconsistent optimization objectives. Recently, end-to-end AD [23, 26, 30, 56] becomes a hot topic, which aims to build a fully differentiable systems for AD so as to enjoy the benefits of data-driven scalable paradigm [55]. Besides, the recent

success of large language models (LLM) has inspired the community to explore the possibility of utilizing the strong common sense within LLM for AD [58].

In this paper, camera calibration emerges as a fundamental task in AD scenarios aimed at ensuring accurate interpretation and understanding of multi-view multimedia images captured by cameras, thereby achieving precise environmental perception, which is critical for ensuring the safety and optimal performance of AD systems.

## 3 METHOD

### 3.1 Preliminaries

*3.1.1 Calibration Parameters.* The calibration parameters of cameras mainly include the intrinsic $\mathbf{I}$ and the extrinsic $\mathbf{E}$ parameters. The intrinsic parameter $\mathbf{I}$ mainly relates to the inherent characteristics of cameras,

$$\mathbf{I} = \begin{bmatrix} f_x & 0 & c_x \\ 0 & f_y & c_y \\ 0 & 0 & 1 \end{bmatrix}, \tag{1}$$

where $f_x$, $f_y$ denote the focal lengths, and $c_x$, $c_y$ denote the coordinates of the principal point. The extrinsic parameter $\mathbf{E}$ is used to transform images from the world coordinate system to the camera coordinate system,

$$\mathbf{E} = \begin{bmatrix} \mathbf{R} & \mathbf{t} \\ \mathbf{0} & 1 \end{bmatrix}, \tag{2}$$

where $\mathbf{R}$ denotes the rotation matrix and $\mathbf{t}$ denotes the translation vector.

*3.1.2 BEV Representations with Calibration Parameters.* Here, we firstly present an overview of the BEV representations [35, 53] and then discuss their close relationship with camera calibration parameters.

The BEV features are defined as a pre-defined grid anchor, portraying the environment as a bird's-eye view, and are represented by a parameter matrix $\mathbf{Q} \in \mathbb{R}^{H \times W \times C}$. The height $H$ and width $W$ denote the region centered around the ego vehicle, and $C$ denote the number of channels in the coordinate region.

To obtain the BEV features of each position, it is necessary to map the multi-view image features to the corresponding anchor positions through the intrinsic and extrinsic calibration parameters [35]. Specifically, suppose that the real-world position $(x', y')$ corresponding to the query $\mathbf{Q}_p \in \mathbb{R}^{1 \times C}$ is located at $p = (x, y)$ of $\mathbf{Q}$,

$$x' = \left( x - \frac{W}{2} \right) \times s, \quad y' = \left( y - \frac{H}{2} \right) \times s, \tag{3}$$

where $s$ denotes the grid size in the real-world corresponding to the BEV features. In addition, it is essential to pre-define a set of anchors $\{z'_j\}_{j=1}^{N_{\text{ref}}}$ to describe the heights along the $z$-axis in the real environment. Finally, we obtain a series of 3D reference points $(x', y', z'_j)$ corresponding to the BEV features $\mathbf{Q}_p$.

According to the calibration parameters in Eqs. 1 and 2, we have the following projection mapping relationships,

$$\mathbf{z}_{ij} \cdot \begin{bmatrix} x_{ij} & y_{ij} & 1 \end{bmatrix}^\top = \mathbf{I}_i \cdot \mathbf{E}_i \cdot \begin{bmatrix} x' & y' & z'_j & 1 \end{bmatrix}^\top, \tag{4}$$

where $(x_{ij}, y_{ij})$ denotes the pixel coordinates of the 3D reference points projected on the $i$-th camera. $\mathbf{I}_i$ and $\mathbf{E}_i$ are the calibration

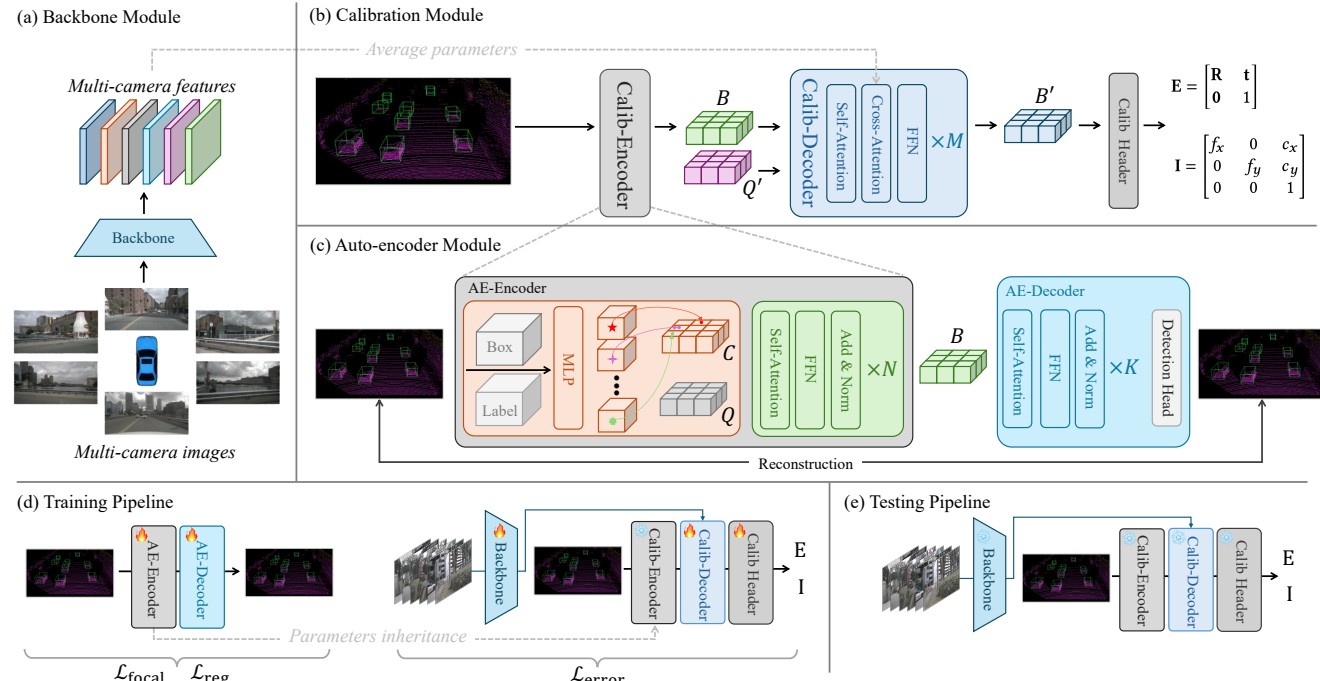

**Figure 2: Overview of our proposed CalibRBEV Model. (a): Multiple images are captured from different perspectives using cameras and then fed into the backbone network to obtain multi-view image features. (b): The calibration module utilizes the bounding boxes and multi-view features as input, which are processed through an attention-based encoder-decoder architecture to refine the reversed BEV representations and predict the calibration parameters. (c): Reconstruction of the auto-encoder module allows for comprehensive exploration of the bounding box information, enabling the pre-acquisition of robust BEV representations. (d): The training pipeline consists of two stages. Note that once the AE-Encoder training is completed, its parameters are frozen and inherited by the Calib-Encoder. Additionally, the Calib-Decoder and AE-Decoder serve distinct purposes: the former predicts calibration parameters, while the latter focuses on reconstruction. (e) During testing stage, given a single-frame multi-view images and bounding box data, we can predict the intrinsic and extrinsic parameters.**

parameters of the $i$-th camera. Note that an additional group of zero-column vectors needs to be included in $\mathbf{I}_i$ to ensure compliance with matrix dimensions. For simplicity, this mapping process is reformulated as follows:

$$\mathcal{P}(p, i, j) = (x_{ij}, y_{ij}). \tag{5}$$

## 3.2 CalibRBEV Model

### 3.2.1 Overview Architecture.
The overall architecture of our proposed CalibRBEV model is shown in Fig. 2, comprises three main components: the backbone network, the predictive calibration parameter module, and auto-encoder reconstruction module. Specifically, initially, the multi-view images captured from multiple cameras are processed through a backbone network to obtain latent feature representations of the surrounding environment. Then, the auto-encoder module is employed to reconstruct 3D bounding boxes, yielding a robust reversed BEV representations, and subsequently preparing for the calibration module. Ultimately, we reverse the entire pipeline of the BEV perception model, which accepts 3D bounding boxes and above multi-view image features as input for attention interaction to predict the camera calibration parameters.

### 3.2.2 Backbone Module.
Initially, we need to extract features from the surround view images. As show in 2(a), given a set of multi-view images $x_{\text{img}} = \{x_{\text{img}}^i\}_{i=1}^{N_{\text{view}}}$ captured by a multi-camera system on a vehicle, a backbone network (e.g. ResNet [21]) is employed to extract a set of multi-view lower-resolution image features $F = \{F^i\}_{i=1}^{N_{\text{view}}}$, where $N_{\text{view}}$ denotes the total number of cameras.

### 3.2.3 Calibration Encoder Module.
The calibration encoder module is utilized simultaneously for predicting calibration parameters and auto-encoder reconstruction. Structurally equivalent, the sole distinction lies in the initialization frozen learnable parameters of the former, derived from pre-training via autoencoder reconstruction.

In order to extract the information from 3D bounding boxes comprehensively, we introduce a bounding box coder initially to enable feature mapping. Formally, we initialize a code map $C$ akin to BEV features as discussed in Sec. 3.1.2 to assimilate information pertaining to the 3D bounding boxes. Suppose that the ground-truth bounding box labels are denoted by $b_{\text{bbox}} = \{b^i\}_{i=1}^M \in \mathbb{R}^9$ and the categorical labels are represented by $c_{\text{bbox}} = \{c^i\}_{i=1}^M \in \mathbb{Z}$, where $M$ denotes the number of objects. Then, we directly concatenate them and pass them through a linear layer to obtain the feature

representation, which is then mapped onto the corresponding code map.

$$C_{p_i} = \text{Linear}([b^i; c^i]), \qquad (6)$$

where $[;]$ denotes the concatenate operation, and $p_i$ refers to the location of the $i$-th object on the code map.

Then, a set of $N$ self-attention blocks is employed to obtain preliminary reversed BEV features $\boldsymbol{B}$. Obviously, as discussed in Sec. 3.1.2, both the BEV features and the multi-view image features entail a considerable number of parameters, leading to substantial computational overhead when directly computing cross-modal attention. In this paper, we build upon the deformable cross-attention [63] instead, which is defined as follows:

$$\text{DeformAttn}(\boldsymbol{q}, \boldsymbol{p}, \boldsymbol{x}) = \sum_{i=1}^{N_{\text{head}}} \boldsymbol{W}_i \sum_{j=1}^{N_{\text{key}}} A_{ij} \cdot \boldsymbol{W}'_i \boldsymbol{x}(\boldsymbol{p} + \Delta \boldsymbol{p}_{ij}), \quad (7)$$

where $\boldsymbol{q}$, $\boldsymbol{p}$, and $\boldsymbol{x}$ denote the query element, 2D reference points, and the input feature map, respectively. $N_{\text{head}}$ and $N_{\text{key}}$ are the total number of attention heads and the sample points. $\Delta \boldsymbol{p}_{ij}$ is the sample offset of the $j$-th sample point in the $i$-th attention head. $A_{ij}$ is the attention weight of the $j$-th sample point in the $i$-th head. $\boldsymbol{W}_i$ and $\boldsymbol{W}'_i$ are learnable weights.

The self-attention mechanism is constructed upon the deformable attention mechanism outlined in Eq. 7, thereby facilitating the generation of BEV features through enhanced interaction with adjacent local information. Moreover, the mechanism entails querying code map features and the reversed BEV query itself.

$$\text{SelfAttn}(\boldsymbol{Q}, p, C) = \text{DeformAttn}(\boldsymbol{Q}_p, p, \boldsymbol{Q}) + \text{DeformAttn}(\boldsymbol{Q}_p, p, C), \qquad (8)$$

where $\boldsymbol{Q}$ denotes the learnable query, $C$ denotes the code map from bounding box as discussed in Eq. 6. The full self-attention block also encompasses both a feed-forward network (FFN) and normalization procedures, ultimately yielding reversed BEV features $\boldsymbol{B}$ after stacking $N$ times.

Particularly, as discussed at the beginning of this subsection, such calibration encoder module undergoes pre-training within the auto-encoder, followed by the seamless transfer of its parameters to the calibration prediction module, where the inherited parameters are entirely frozen to preserve the capability for reversed BEV feature extraction. We finally summarize the above reversed BEV feature extraction process as follows:

$$\boldsymbol{B} = \text{CalibEncoder}(\boldsymbol{Q}, b_{\text{bbox}}, c_{\text{bbox}}). \qquad (9)$$

### 3.2.4 Calibration Decoder Module.
Once the reversed BEV features are obtained, the calibration decoder module takes in a novel learnable BEV queries $\boldsymbol{Q}'$ and the reversed BEV features $\boldsymbol{B}$, then merges with multi-view image features $\{F^i\}_{i=1}^{N_{\text{view}}}$ from backbone network, as discussed in Sec. 3.2.2, ultimately yielding the enhanced reversed BEV feature $\boldsymbol{B}'$ to predict the calibration parameters.

In detail, the space surrounding the vehicle is divided into $H \times W$ cells, with each cell corresponding to a BEV feature. Each BEV feature only utilizes the approximate position of the corresponding 3D space projected onto a 2D image as a reference point. Concurrently, we also require interaction with multi-view image features through

multi-view cameras. Finally, the cross-attention mechanism is formulated as follows:

$$\text{CrAttn}(\boldsymbol{Q}'_p, F^i) = \frac{1}{|\mathcal{V}_{\text{hit}}|} \sum_{i \in \mathcal{V}_{\text{hit}}} \sum_{j=1}^{N_{\text{ref}}} \text{DeformAttn}(\boldsymbol{Q}'_p, \mathcal{P}(p, i, j), F^i), \qquad (10)$$

where $\mathcal{V}_{\text{hit}}$ denotes the set of image features mapped correspondingly to the BEV features $\mathcal{P}(p, i, j)$ according to Eqs. 4 and 5. $F^i$ denotes the $i$-th camera image feature as discussed in Sec. 3.2.2. Here, the mapping relationship in attention mechanism is established directly using uniform average calibration parameters. Overall, in the calibration decoder module, there are concurrent implementations of self-attention in Eq. 8 and cross-attention in Eq. 10, alongside a FFN. This whole attention module undergoes $M$ iterations of stacking, eventually yielding refined reversed BEV feature $\boldsymbol{B}'$.

Following this, the refined BEV feature $\boldsymbol{B}'$ is further subjected to a conventional transfromer decoder process within which the target feature $\boldsymbol{T}$ is generated. The target feature $\boldsymbol{T}$ is utilized across three distinct fully connection branches, each serving to predict the calibration parameters.

$$\begin{aligned} \mathbf{R}' &= \text{Linear}_{\mathbf{R}}(\boldsymbol{T}), \\ \mathbf{t}' &= \text{Linear}_{\mathbf{t}}(\boldsymbol{T}), \\ \mathbf{I}' &= \text{Linear}_{\mathbf{I}}(\boldsymbol{T}). \end{aligned} \qquad (11)$$

We summarize above calibration decoder and final prediction header as follows.

$$\boldsymbol{B}' = \text{CalibDecoder}(\boldsymbol{Q}', \boldsymbol{B}), \qquad (12)$$

$$\mathbf{E}', \mathbf{I}' = \text{CalibHeader}(\boldsymbol{B}'). \qquad (13)$$

### 3.2.5 Auto-encoder Module.
In order to proactively obtain robust reversed BEV representations, we propose employing an attention-based auto-encoder's reconstruction strategy for pre-training, as shown in Fig. 2(c). For the encoder component within the autoencoder architecture, it adopts the same architecture as calibration encoder in Eq. 9.

$$\boldsymbol{B} = \text{AEEncoder}(\boldsymbol{Q}, b_{\text{bbox}}, c_{\text{bbox}}) = \text{CalibEncoder}(\boldsymbol{Q}, b_{\text{bbox}}, c_{\text{bbox}}). \qquad (14)$$

For the decoder component, the reversed BEV features $\boldsymbol{B}$ are supposed to output to a regular attention object detection decoder to generate 3D object bounding boxes again. We apply a similar self-attention mechanism in Eq. 8 and stack $K$ layers to obtain the final output of the 3D bounding boxes.

$$b'_{\text{bbox}}, c'_{\text{bbox}} = \text{AEDecoder}(\boldsymbol{B}). \qquad (15)$$

### 3.2.6 Loss Functions.
Our training procedure consists of two stages as shown in Fig. 2(d). (1) reconstruction pre-training of the auto-encoder module and (2) train the calibration module while freezing the encoder module. For the former, focal loss and L1 loss are used for classification and bounding box regression, respectively.

$$\mathcal{L}_{\text{focal}} = -\alpha(1 - p_{c'_{\text{bbox}}})^{\gamma} \log(p_{c'_{\text{bbox}}}), \qquad (16)$$

$$\mathcal{L}_{\text{reg}} = \|b'_{\text{bbox}} - b_{\text{bbox}}\|_1, \qquad (17)$$

where $p_{c'_{\text{bbox}}}$ denotes the estimated probability of category $c'_{\text{bbox}}$. $\alpha$ and $\gamma$ are the hyper-parameters. For the latter, all the calibration

**Table 1: Comparison results of parameters prediction errors on nuScenes dataset. CalibRBEV-$N$ refers to our model trained with $N$ cameras dataset. Here, t, R, and I denote the distance error of translation vector, angle error of rotation matrix, and error percentage of intrinsic parameters, respectively. $\bar{f}$ and $\bar{c}$ are the average error percentage of the focal length and the principal point. The symbol ↓ indicates that lower values represent better performance.**

| Method | $\mathbf{t}$ $(m)$ ↓ | $\mathbf{R}$ $(°)$ ↓ | $\mathbf{I}$ $(\%)$ ↓ | $\bar{f}$ $(\%)$ ↓ | $\bar{c}$ $(\%)$ ↓ |
|---|---|---|---|---|---|
| DeepPTZ [60] | - | - | - | 0.0468 | 0.0850 |
| DeepCalib [3] | - | - | - | 0.1517 | 0.1790 |
| DeepHome-DS | - | - | - | 0.1426 | 0.1740 |
| LCCNet [42] | 0.0242 | 0.4234 | - | - | - |
| CalibRBEV-(300 cameras) | **0.0102** | 0.1637 | 0.0287 | 0.0245 | 0.0329 |
| CalibRBEV-(4200 cameras) | 0.0193 | **0.0614** | **0.0188** | **0.0190** | **0.0185** |

**Table 2: Comparison results of parameters prediction errors on Waymo dataset. CailbRBEV-org denotes a model trained on the original Waymo dataset. The symbol ↓ indicates that lower values represent better performance.**

| Method | $\mathbf{t}$ $(m)$ ↓ | $\mathbf{R}$ $(°)$ ↓ | $\mathbf{I}$ $(\%)$ ↓ | $\mathbf{t}_x$ $(m)$ ↓ | $\mathbf{t}_y$ $(m)$ ↓ | $\mathbf{t}_z$ $(m)$ ↓ |
|---|---|---|---|---|---|---|
| Edge-Extraction [59] | - | - | - | 0.0890 | 0.0920 | 0.0730 |
| SSI [64] | - | - | - | 0.0930 | 0.0750 | 0.0870 |
| Coarse [46] | - | - | - | 0.1210 | 0.1890 | 0.2020 |
| IS [46] | - | - | - | 0.0480 | 0.0290 | 0.0410 |
| CailbRBEV | 0.0048 | **1.2762** | 1.2630 | 0.0031 | 0.0026 | 0.0010 |
| CailbRBEV-org | **0.0016** | 1.5783 | **0.9992** | **0.0010** | **0.0009** | **0.0003** |

errors of the intrinsic and extrinsic parameters are used for training, where the errors are calculated by L1 loss.

$$\mathcal{L}_{\text{calib-error}} = \|\mathbf{E}' - \mathbf{E}\|_1 + \|\mathbf{I}' - \mathbf{I}\|_1, \quad (18)$$

where $\|\cdot\|_1$ here denotes the matrix element-wise norm, and the rotation matrix is represented using quaternions.

*3.2.7 Summary.* Once the model training is complete, given a single-frame multi-view images and on-board bounding box from LiDAR sensor input, we can predict the intrinsic and extrinsic parameters using the calibration prediction module as shown in Fig. 2(e). This process eliminates the need for additional complex manual procedures or multi-frame inputs, significantly alleviating the cost burden of calibration tasks. Furthermore, a potential avenue for extension lies in the real-time implementation of our approach directly within on-board vehicles, thereby significantly enhancing its applicability and utility in AD scenarios.

## 4 EXPERIMENTAL RESULTS

### 4.1 Dataset

We conduct experiments on nuScenes [5] and Waymo [47] datasets in this paper for calibration parameters evaluation.

*4.1.1 The nuScenes Dataset.* The nuScenes dataset consists of $1,000$ sequences, each spanning approximately 20 seconds and captured at a sampling rate of 20 frames per second, where the key samples are annotated at 2Hz. In total, there exist $28k$, $6k$, and $6k$ annotated instances for training, validation, and testing, respectively. In addition, each sample contains RGB images from 6 cameras: `front_left`, `front`, `front_right`, `back_left`, `back`, `back_right`.

Hence, the nuScenes dataset encompasses a total of 6000 calibrated cameras. Among these, the subset of 900 cameras is withheld from public access, designated as the challenge test dataset, while an additional 900 cameras are specifically used for validation. And the remaining 4200 cameras are used for training.

*4.1.2 The Waymo Open Dataset.* As for the Waymo dataset, in total, there are 1150 sequences in the dataset, encompassing 798 training sequences, 202 validation sequences, and 150 test sequences. Each sample is captured by 5 cameras, consisting of three frontal views and two side views, denoted as `side_left`, `front_left`, `front`, `front_right`, and `side_right`. In practice, data from 90 vehicles are designated as the training set, while data from 25 vehicles are allocated for validation. Each sequence spans 20 seconds, and at intervals of every 5 frames, up to 200 frames are randomly sampled for training.

*4.1.3 Evaluation Metrics.* The error in calibration parameters is utilized as the primary evaluation metric. Moreover, the 3D object detection metric [35], such as mAP, mATE, mASE, mAOE, etc, are also used for comprehensive evaluation.

### 4.2 Implementation Details

The ResNet101-DCN [21] pre-trained with FCOS3D [51] is implemented as the backbone module. For the reversed BEV representations, we encapsulate the real environment within a 100 meter radius around the vehicle to generate features with dimensions (height $H$, width $W$, number of channels $C$) of (150, 150, 256). For the training of auto-encoder module, the AdamW optimizer is adopted with a weight decay of 0.01, the learning rate is initially set to 0.0002

**Table 3: Ablation results of our module architecture design on nuScenes dataset with** 4200 **cameras data.**

| # | Reversed Pipe | AE-Finetune | AE-Frozen | t (m) ↓ | R (°) ↓ | I (%) ↓ | $f_x$ (%) ↓ | $f_y$ (%) ↓ | $c_x$ (%) ↓ | $c_y$ (%) ↓ |
|---|---|---|---|---|---|---|---|---|---|---|
| 1 | ✓ | | | 0.0281 | 0.0790 | 0.0207 | 0.0198 | 0.0203 | 0.0208 | 0.0220 |
| 2 | ✓ | ✓ | | 0.0315 | 0.1087 | 0.0233 | 0.0209 | 0.0221 | 0.0221 | 0.0282 |
| 3 | ✓ | | ✓ | **0.0193** | **0.0614** | **0.0188** | **0.0188** | **0.0192** | **0.0191** | **0.0180** |

**Table 4: Ablation results of our module architecture design on nuScenes dataset with** 300 **cameras data.**

| # | Reversed Pipe | AE-Finetune | AE-Frozen | t (m) ↓ | R (°) ↓ | I (%) ↓ | $f_x$ (%) ↓ | $f_y$ (%) ↓ | $c_x$ (%) ↓ | $c_y$ (%) ↓ |
|---|---|---|---|---|---|---|---|---|---|---|
| 1 | ✓ | | | 0.0254 | 0.1355 | 0.0455 | 0.0390 | 0.0374 | 0.0442 | 0.0615 |
| 2 | ✓ | ✓ | | 0.0323 | **0.1300** | 0.0439 | 0.0350 | 0.0323 | 0.0404 | 0.0678 |
| 3 | ✓ | | ✓ | **0.0102** | 0.1637 | **0.0287** | **0.0260** | **0.0230** | **0.0256** | **0.0403** |

**Table 5: Performance comparisons of the detection on BEVFormer-S and PETRv2-vov using the calibration parameters derived from our CalibRBEV model. CalibRBEV-**N **refers to our model trained with** N **cameras dataset.**

| Model | Method | mAP ↑ | NDS ↑ | mATE ↓ | mASE ↓ | mAOE ↓ | mAVE ↓ | mAAE ↓ |
|---|---|---|---|---|---|---|---|---|
| | Trad. Calib. [61] | 0.366 | 0.480 | 0.729 | 0.278 | 0.371 | 0.446 | 0.201 |
| BEVFormer-S | CalibRBEV-(4200 cameras) | 0.363 | 0.478 | 0.739 | 0.278 | 0.367 | 0.450 | 0.201 |
| | CalibRBEV-(300 cameras) | 0.363 | 0.477 | 0.742 | 0.278 | 0.370 | 0.450 | 0.201 |
| | Trad. Calib. [61] | 0.410 | 0.502 | 0.723 | 0.269 | 0.454 | 0.390 | 0.193 |
| PETRv2-vov | CalibRBEV-(4200 cameras) | 0.409 | 0.497 | 0.725 | 0.270 | 0.488 | 0.398 | 0.191 |
| | CalibRBEV-(300 cameras) | 0.400 | 0.495 | 0.752 | 0.270 | 0.446 | 0.393 | 0.186 |

and decayed with a cosine annealing. To train the calibration module, the same optimizer schedule as that of the auto-encoder module is employed, though with an adjustment in the learning rate specifically for the image backbone, which is lowered to 0.00002. It is noteworthy that the two modules are trained independently, as depicted in Figure 2(d). The parameters of the calibration encoder module are inherited from the auto-encoder encoder, and they remain frozen during the overall calibration module training process.

## 4.3 Main Results

Table 1 presents the comparative experimental results on nuScenes dataset, including errors in translation vector **t**, rotation matrix **R**, intrinsic parameters **I**, focal length $f$, and principal point $c$. The findings reveal that our model consistently exhibits outstanding prediction performance. As shown in Table 1, when model is trained with total 4200 cameras data, our model achieves 0.0614 and 0.0188 error on rotation matrix and intrinsic parameters, respectively. Furthermore, as the calibration training data is reduced, it still achieves a translation vector error of 0.0102, demonstrating generalization capabilities of our model.

As for the comparison results on the Waymo dataset in Table 2, we attained errors of 0.0016, 1.276, and 1.2630 in translation vector, rotation matrix, and intrinsic parameters, respectively, serve as evidence validating the effectiveness of the model. Subsequently, further comparison is conducted regarding the errors in translation vectors along the XYZ axes. The performance of our model in predicting translation vectors is significantly improved, with

errors reduced to 0.0031, 0.0026, and 0.0010, outperforming previous results of 0.0480, 0.0290, and 0.0410, respectively. Performance is further enhanced upon employing the original Waymo dataset, resulting in errors reduced to 0.001, 0.0009, and 0.0003.

## 4.4 Ablation Study

Table 3 and 4 present the results of our ablation results on the nuScenes dataset. The former is based on a 4200 cameras dataset, while the latter experimented solely with 300 cameras dataset. The "Reversed Pipe" is an abbreviation for our overall reversed BEV pipeline, indicating the direct training of the calibration module to achieve the prediction of calibration parameters. "AE-Finetune" and "AE-Frozen" signify the incorporation of auto-encoder pretraining, representing parameter fine-tuning and parameter freezing, respectively. Clearly, when utilizing the reversed BEV pipeline along with a frozen AE, we observe a notable overall performance improvement on both different dataset size, validating the effectiveness of our architectural design. Additionally, an increase in data volume contributes to more accurate predictive outcomes. Finally, particularly concerning intrinsic parameters, the performance of the frozen AE improved from 0.0207 to 0.0188 ($\Delta$ = 0.0019) in the case of the 4200 dataset. And in the scenario of only 300 cameras dataset, the performance improved significantly from 0.0455 to 0.0287 ($\Delta$ = 0.0168), providing further validation of the AE's performance on small calibration dataset.

**Table 6: Performance comparisons of the detection using auto-encoder module of our CalibRBEV model.**

| Method | mAP ↑ | NDS ↑ | mATE ↓ | mASE ↓ | mAOE ↓ | mAVE ↓ | mAAE ↓ |
|---|---|---|---|---|---|---|---|
| FCOS3D [51] | 0.358 | 0.428 | 0.690 | 0.249 | 0.452 | 1.434 | 0.124 |
| DETR3D [52] | 0.412 | 0.479 | 0.641 | 0.255 | 0.394 | 0.845 | 0.133 |
| BEVDet4D [24] | 0.451 | 0.569 | 0.511 | 0.241 | 0.386 | 0.301 | 0.121 |
| BEVFormer [35] | 0.481 | 0.569 | 0.582 | 0.256 | 0.375 | 0.378 | 0.126 |
| PETRv2 [39] | 0.490 | 0.582 | 0.561 | 0.243 | 0.361 | 0.343 | **0.120** |
| BEVDepth [34] | 0.503 | 0.600 | 0.445 | 0.245 | 0.378 | 0.320 | 0.126 |
| CalibRBEV (Auto-Encoder) | **0.824** | **0.842** | **0.247** | **0.077** | **0.105** | **0.092** | 0.182 |

## 4.5 Further Exploration

We additionally assesses the calibration parameters obtained by our CalibRBEV model in other object detection models, replacing the traditional calibration's intrinsic and extrinsic parameters [5, 61] with the parameters we predicted.

We adopt BEVFormer-S [35] and PETRv2-vov [39] as our baseline methods. Specifically, the BEVFormer-S model directly utilizes weights trained on the original dataset, while PETRv2-vov is trained using our predicted calibrated intrinsic and extrinsic parameters. As shown in Table 5, compared to traditional methods that are complex and time-consuming, utilizing the predicted calibration parameters for detection yields very close results, indicating that the model can be integrated into other BEV methods without performance degradation. Particularly, using only 300 cameras dataset, applying our model's prediction parameters to the object detection models, still yielding approximate results, confirming the efficacy of predicting calibration parameters with limited data.

Moreover, we also assesses the detection performance results of the auto-encoder module within our CalibRBEV model in Table 6. Obviously, the auto-encoder module demonstrates outstanding detection capabilities, suggesting that the generated BEV features encompass sufficient 3D object bounding box information and satisfy the requirements of the calibration module.

## 5 DISCUSSION

In this section, we discuss the limitations of our work and prospective directions for future research. We hope that our work to serve as a catalyst inspiring further future endeavors.

(1) Firstly, in this paper, we extract multi-view features from multi-camera images and directly obtain 3D bounding boxes from LiDAR to achieve a reversed BEV pipeline for calibration parameters prediction. However, the information relationships between multiple sensors and fusion strategies merit further exploration to enhance the performance of parameter prediction.

(2) Then, our current model still relies on predicting calibration parameters corresponding to single-frame data. However, in the context of autonomous driving scenarios, spatiotemporal information is accessible. Learning the correlation of the overall spatiotemporal structure can endow networks with structural knowledge derived from motion, thereby potentially enhancing the performance of calibration tasks.

(3) Furthermore, the mapping relationship between the attention interaction mechanism of image features and BEV (Bird's Eye View)

features, which is pre-determined using average parameters, may possess potential inaccuracies. Thus, there could be a necessity to devise a new attention interaction architecture in the future.

(4) Finally, our model falls within the realm of learning-based calibration methods, substantially reducing traditional calibration costs. However, akin to other deep learning models, it necessitates pre-existing datasets for training to enable parameter prediction. Although this work introduces an auto-encoder module to alleviate the demand for extensive datasets, a potential future direction is to explore methods for calibrating parameter prediction using fewer training data.

## 6 CONCLUSION

In this paper, we introduce a novel multi-camera calibration method for calibrating intrinsic and extrinsic parameters, termed CalibRBEV, which innovatively proposes the use of a reverse pipeline based on the BEV model. Our model mainly consists of two components: the calibration module and the auto-encoder module. The auto-encoder module fully explores latent environmental information to achieve the extraction of reversed BEV features, and the calibration module then predicts calibration parameters leveraging multi-view and reversed BEV features. Experimental results demonstrate superior performance compared to previous calibration methods. Furthermore, the model can be extended to scenarios with small data volumes, simplifying the requirements of extensive calibration datasets and thereby optimizing calibration efforts.

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
