# OpenReview forum: "CalibRBEV: Multi-Camera Calibration via Reversed Bird's-eye-view Representations for Autonomous Driving"
_acmmm.org/ACMMM/2024/Conference — MM2024 Poster_

### Official Review · Reviewer_7EhL · 2024-05-20

**Rating:** 2
**Confidence:** 3

**Summary:**

This paper reverses the BEV 3D object detection process to predict the calibration parameters with multi-camera images and Lidar 3D object bounding boxes as input. The proposed CalibRBEV model comprises two key modules: the calibration module and the auto-encoder module. The auto-encoder module explores latent environmental information to achieve the extraction of reversed BEV features, and the calibration module predicts calibration parameters by leveraging multi-view and reversed BEV features.

**Strengths:**

The idea and explored task in this paper is quite new.
The paper is well-written and easy to follow.

**Limitations:**

Although the task is quite new, the approach of using both multi-camera images and LiDAR 3D object bounding boxes as input makes it less practical. The effort required to obtain 3D object bounding boxes is significantly higher than that needed for camera calibration, making the task setting questionable.

Has the author compared the time and computational costs with traditional calibration methods? Efficiency and cost are crucial factors in calibration.

Has the author considered using only image data as input? Achieving calibration with image-only data would be a more reasonable and practical approach.

**Suitability:**

3

---

### Official Review · Reviewer_eZ7a · 2024-05-24

**Rating:** 3
**Confidence:** 3

**Summary:**

This manuscript proposes CalibRBEV, a multi-camera calibration method designed for autonomous driving. The calibration module takes the multi-view image features and bounding boxes as input. The method utilizes an attention autoencoder module to reconstruct bounding boxes in order to extract the reversed BEV representations. Then, the obtained representations are combined with multi-view image features to facilitate the prediction of camera calibration parameters. According to the manuscript, the CalibRBEV method can achieve efficient and robust multi-camera calibration in autonomous driving scenarios, demonstrated in nuScenes and Waymo datasets.

**Strengths:**

1.	By introducing the concept of BEV perception into the calibration task, the proposed CalibRBEV provides a new idea to solve the problem of multi-camera camera calibration in autonomous driving.

2.	This manuscript innovatively proposes a pre-training strategy based on attention autoencoder module. It aims to extract high-quality BEV representations by reconstructing 3D object bounding box, thereby enhancing the prediction of calibration parameters.

3.	During training pipeline, it has a two-stage strategy. The autoencoder module is optimized through reconstruction pre-training; then, the calibration module is trained while freezing the encoder module. This facilitates the model to learn feature representation and subsequently improve prediction of calibration parameters.

4.	The experimental design is adequate. It demonstrates that CalibRBEV achieves good performance compared to other methods on both the nuScenes and Waymo datasets. Ablation studies also provide evidence that the autoencoder pre-training contributes to the performance improvement. Furthermore, the authors apply the predicted calibration parameters to BEVFormer-S and PETRv2-vov, validating the effectiveness and compatibility of CalibRBEV with existing BEV perception frameworks. Overall, the experimental results show the effectiveness of the proposed CalibRBEV method in multi-camera calibration for autonomous driving scenarios.

**Limitations:**

This manuscript can be further enhanced by carefully considering the following comments:
1.	There are duplicate and redundant contents in the article, making it appear overly lengthy. For instance, the first sentence of the Abstract:" Camera calibration consists of determining the intrinsic and extrinsic parameters of an imaging system, which forms the fundamental basis for various computer vision tasks and applications." And the first sentence of the Introduction:" Camera calibration is a fundamental procedure in computer vision tasks [43, 52], mainly involving the determination of intrinsic and extrinsic parameters of an imaging system.". I think the manuscript needs to be revised, especially Introduction section.

2.	There are some grammatical mistakes in the manuscript. To enhance the readability and professionalism of the paper, the authors should conduct a further examination. Here are the grammar problems I noticed:
	In the Abstract, 'we proposes' should be corrected to 'we propose'.
	At line 393-395, “…model is shown in Fig. 2, comprises three main components:” contains a grammatical error.
	At line 827 and line 843, “We additionally assesses the calibration…” and “…we also assesses the detection…” both contain grammatical errors.

3.	At line 131 of the manuscript, the "auto-encoder reconstruction strategy" is mentioned as being able to capture environmental information and contribute to accurate prediction of camera calibration parameters. However, there is no specific explanation provided on why this strategy are adopted. The authors need to supplement relevant information.

4.	In the Abstract and Introduction, authors point out the time-consuming problem of traditional/prevailing camera calibration methods. However, experimental results do not contain the runtime performance and cost of the proposed CalibRBEV. It would be better if the authors could supplement this information.

**Suitability:**

2

---

### Official Review · Reviewer_QVkX · 2024-05-25

**Rating:** 3
**Confidence:** 4

**Summary:**

This submission has a very novel idea: predicting the camera intrinsic and extrinsic parameters from input RGB images and ground truth bounding boxes. The ground truth boxes are encoded by an auto-encoder into features, and cross-attended with input RGB image features and finally used to predict intrinsic and extrinsic camera parameters. Quantitative experiments are presented and nuScenes and Waymo, showing improved camera parameter prediction results compared with baselines. The authors also show good box auto-encoding results in table.6.

**Strengths:**

+ The idea is definitely novel and I don't remember anyone doing this before: predicting camera intrinsic and extrinsic camera parameters from the cross-attened features between input RGB images and ground truth bounding boxes.
+ Doing bounding box auto-encoding boils down to a generative model that captures the joint distribution of outdoor scene layout. While this methodology has been used for indoor scene layout synthesis for a long time, this is the first time I see this for autonomous driving. According to the results in Table.6, the results are quite strong so this module may be useful in related downstream tasks, not specifically this one.

**Limitations:**

- The biggest issue is the setting. While what is done here is novel, I cannot agree this is a calibration process. The authors exploit representations extracted from ground truth bounding boxes as input, but this is not doable actually in the real world. When we are trully faced with a calibration challenge, how could we get these bounding boxes? By contrast, I think a V2X setting would be more interesting. The authors could argue that the road side box detections are accurate thus can be used as input into the parameter prediction network.
- Scene layout autoencoding can be used as a generative prior for detection tasks, instead of this parameter prediction task. There are other interesting downstream task considering the calibration setting does not stand actually, IMHO.

**Suitability:**

2

---

### Meta-Review · Area_Chair_GBSs · 2024-07-02

**Recommendation:** Accept (Poster)
**Confidence:** 4

**Metareview:**

The paper receives mixed ratings from three reviewers after the rebuttal. Two reviewers recommend borderline accept and one weak reject.  The two reviewers with positive ratings both like the idea and are convinced by the extensive experimental evaluation. The last reviewer feels that the setting used in the paper is questionable as it requires both bounding boxes and multi-view images as input for the camera calibration, however, this is indeed a novelty of the method itself, as acknowledged by the other two reviewers. AC also thinks that the proposed method is interesting and shows promising results, and thus recommends acceptance for this time.